



# A bias-corrected GEMS geostationary satellite product for nitrogen dioxide using machine learning to enforce consistency with the TROPOMI satellite instrument

Yujin J. Oak[1], Daniel J. Jacob[1,2], Nicholas Balasus[1], Laura H. Yang[1], Heesung Chong[3], Junsung Park[3],
Hanlim Lee[4], Gitaek T. Lee[5], Eunjo S. Ha[5], Rokjin J. Park[5], Hyeong-Ahn Kwon[6], Jhoon Kim[7]

[1]School of Engineering and Applied Sciences, Harvard University, Cambridge, MA, USA
[2]Department of Earth and Planetary Sciences, Harvard University, Cambridge, MA, USA
[3]Harvard-Smithsonian Center for Astrophysics, Cambridge, MA, USA
[4]Division of Earth Environmental System Science, Major of Spatial Information Engineering, Pukyong National University,
Busan, South Korea
[5]School of Earth and Environmental Science, Seoul National University, Seoul, South Korea
[6]Department of Environmental and Energy Engineering, University of Suwon, Suwon, South Korea
[7]Department of Atmospheric Sciences, Yonsei University, Seoul, South Korea

*Correspondence to*: Yujin J. Oak (yjoak@g.harvard.edu)

**Abstract.** The Geostationary Environment Monitoring Spectrometer (GEMS) launched in February 2020 is now providing
continuous daytime hourly observations of nitrogen dioxide ($NO_2$) columns over East Asia (5°S–45°N, 75°E–145°E) with
$3.5 \times 7.7$ km$^2$ pixel resolution. These data provide unique information to improve understanding of the sources, chemistry,
and transport of nitrogen oxides ($NO_x$) with implications for atmospheric chemistry and air quality, but opportunities for
direct validation are very limited. Here we correct the operational level-2 (L2) $NO_2$ vertical column densities (VCDs) from
GEMS with a machine learning (ML) model to match the much sparser but more mature observations from the low Earth
orbit TROPOspheric Monitoring Instrument (TROPOMI), preserving the data density of GEMS but making them consistent
with TROPOMI. We first reprocess the GEMS and TROPOMI operational L2 products to use common prior vertical $NO_2$
profiles (shape factors) from the GEOS-Chem chemical transport model. This removes a major inconsistency between the
two satellite products and greatly improves their agreement with ground-based Pandora $NO_2$ VCD data in source regions.
We then apply the ML model to correct the remaining differences, Δ(GEMS-TROPOMI), using as predictor variables the
GEMS $NO_2$ VCDs and retrieval parameters. We train the ML model with collocated GEMS and TROPOMI $NO_2$ VCDs,
taking advantage of TROPOMI off-track viewing to cover a wide range of effective zenith angles (EZAs) for the GEMS
diurnal profiles. The two most important predictor variables for Δ(GEMS-TROPOMI) are GEMS $NO_2$ VCD and EZA. The
corrected GEMS product is unbiased relative to TROPOMI and shows a diurnal variation over source regions more
consistent with Pandora than the operational product.



## 1 Introduction

Nitrogen oxides ($NO_x \equiv NO + NO_2$) are reactive trace gases emitted from combustion, lightning, and microbial activity in soils. Emission is mainly as nitrogen oxide (NO) which cycles rapidly with nitrogen dioxide ($NO_2$). This cycling produces tropospheric ozone ($O_3$) and oxidation of $NO_x$ produces nitrate particulate matter (PM), with implications for air quality, climate forcing, and nitrogen deposition. Satellite measurements of $NO_2$ vertical column densities (VCDs) by solar backscatter from polar sun-synchronous low Earth orbit (LEO) have been used extensively to monitor $NO_x$ emissions and their trends worldwide (Martin et al., 2003; Lamsal et al., 2015; Curier et al., 2014; Duncan et al., 2016; Liu et al., 2017) and to improve understanding of $NO_x$ oxidation chemistry (Boersma et al., 2008; Valin et al., 2013; Miyazaki et al., 2017; Beirle and Wagner, 2024).

$NO_2$ has been measured continuously from LEO since 1995, starting with the Global Ozone Monitoring Experiment (GOME) (Burrows et al., 1999) followed by the operational GOME-2 series (Munro et al., 2016). The Ozone Monitoring Instrument (OMI) was launched in 2004 and is providing global daily continuous data at $13 \times 24$ km$^2$ nadir resolution (Levelt et al., 2006). The TROPOspheric Monitoring Instrument (TROPOMI) launched onboard the Sentinel-5 Precursor (S5P) satellite in 2017 improved the resolution to $3.5 \times 5.5$ km$^2$ (Veefkind et al., 2012).

OMI and TROPOMI retrievals of $NO_2$ have been extensively validated using ground-based measurements of $NO_2$ VCDs from sun-staring Pandora spectrometers and Multi-Axis Differential Optical Absorption Spectroscopy (MAX-DOAS) instruments, and also by intercomparisons with each other (Herman et al., 2019; Pinardi et al., 2020; Wang et al., 2020; Cai et al., 2022; Gu et al., 2023). TROPOMI $NO_2$ is routinely validated against Pandora, MAX-DOAS, and OMI by the Royal Netherlands Meteorological Institute (KNMI), showing overall good agreement with a $-7\%$ mean bias (Lambert et al., 2023). It has been used to quantify $NO_x$ emissions (Goldberg et al., 2019), infer surface $NO_2$ concentrations (Cooper et al., 2020), and evaluate air quality models (Douros et al., 2023).

A limitation with polar sun-synchronous LEO satellites is that they observe a given location at most once per day and at the same time of day. Geostationary satellites can observe at much higher frequency and in principle continuously, providing much denser data and a unique capability for tracking the diurnal cycle of emissions, oxidation chemistry, and pollutant transport. The Geostationary Environment Monitoring Spectrometer (GEMS) was launched onboard the Korea Aerospace Research Institute GEO-KOMPSAT2B (GK2B) satellite in February 2020, in an equatorial plane at 128.2°-longitude with continuous view of East Asia at $3.5 \times 7.7$ km$^2$ pixel resolution over Korea (Kim et al., 2020). It is now providing the first $NO_2$ observations from geostationary orbit. GEMS is part of a geostationary air quality constellation to include TEMPO over North America launched in April 2023 (Zoogman et al., 2017) and Sentinel-4 over Europe to be launched in 2025 (Ingmann et al., 2012).

As with all satellite observations, GEMS retrievals take some time to mature. $NO_2$ VCDs (L2 products) are currently retrieved operationally with the version 2.0 algorithm (National Institute of Environmental Research, 2020). Evaluation of the GEMS $NO_2$ product with four urban Pandora observations in South Korea found an underestimate and disagreements in



diurnal patterns (Kim et al., 2023). However, the local Pandora data may not be representative of the 3.5 × 7.7 km² GEMS
pixels particularly in urban environments. GEMS evaluation for China found both high and low biases in comparison with
ground-based observations and other satellite products (Li et al., 2023; Zhang et al., 2023). Yang et al. (2024) reprocessed
the GEMS NO₂ version 2.0 data with prior NO₂ vertical profiles from the GEOS-Chem chemical transport model (CTM) on
a 0.25° × 0.3125° grid and found good agreement with Pandora in Seoul and Beijing including diurnal variations.

Here we introduce a bias-corrected GEMS NO₂ product using machine learning (ML) to minimize the biases between
GEMS and TROPOMI, and provide in this manner a more reliable and consistent satellite product for scientific applications
and for improving the GEMS retrieval. Although TROPOMI observes over a limited range of solar zenith angles (SZAs) on
account of the overpass time at approximately 13:30 local time (LT), it has off-track viewing to ±50° providing a range of
viewing zenith angles (VZAs) to mimic the wider range of SZAs seen by GEMS. The bias-corrected GEMS product
combines the high data density of GEMS with the accuracy of TROPOMI demonstrated by extensive validation and
algorithm development (Gu et al., 2023; Eskes et al., 2022). Our approach is as follows. First, we adjust for biases caused by
different prior information used in the operational L2 retrievals by applying common prior vertical profiles simulated by
GEOS-Chem. Second, we train an ML model using GEMS and TROPOMI NO₂ VCDs to minimize differences in collocated
data during 2022–2023 using GEMS retrieval parameters as explanatory variables. Third, we apply the trained ML model to
the ensemble of GEMS data for July 2022–June 2023, producing a bias-corrected GEMS product which has the temporal
coverage of GEMS and is consistent with TROPOMI. We use the ML model to identify the GEMS retrieval parameters
associated with the largest discrepancies with TROPOMI and validate this bias-corrected GEMS product with ground-based
observations from the Pandora spectrometers.

## 2 GEMS, TROPOMI, and Pandora operational products

GEMS is an ultraviolet (UV)-visible (VIS) hyperspectral imaging spectrometer that measures solar backscatter with 0.6 nm
spectral resolution over East Asia (5−45° latitude, 75−145° longitude) with 3.5 × 7.7 km² pixels at 37.5°-latitude and hourly
repeat times (08:45−17:45 LT) (Kim et al., 2020). Total NO₂ slant column densities (SCDs) along the sun-satellite light path
are retrieved using differential optical absorption spectroscopy (DOAS) (Platt, 1994) with a fitting window of 435–450 nm
(National Institute of Environmental Research 2020b; Kim et al., 2020). We use the operational L2 product from version 2.0
(available online: https://nesc.nier.go.kr/en/html/index.do, last access: 19 January 2024).
TROPOMI is similarly a UV-VIS hyperspectral imaging spectrometer with a full width at half maximum (FWHM) of 0.55
nm. It is a push-broom instrument with a wide swath of ~2600 km and an off-track viewing angle of ±50° off-nadir,
providing global daily viewing once a day at approximately 13:30 LT with 5.5 × 3.5 km² nadir spatial resolution. Total NO₂
SCDs are retrieved using DOAS in the 405–465 nm window (van Geffen et al., 2022). Here we use the offline (OFFL) L2
product produced by the KNMI NO₂ processor version 2.4.0 (available online: https://dataspace.copernicus.eu/explore-
data/data-collections/sentinel-data/sentinel-5p, last access: 19 January 2024).



Ground-based NO$_2$ column measurements from Pandora instruments have been widely used to validate satellite retrievals (Park et al., 2022; Kim et al., 2023; Pinardi et al., 2020; Yang et al., 2024). Pandora spectrometers provide direct-sun observations to retrieve ground-based NO$_2$ VCDs by DOAS using a spectral fitting window of 400–470 nm (Herman et al., 2009). Here we use the L2 total NO$_2$ VCDs processed by the BlickP software (available online: https://www.pandonia-global-network.org, last access: 19 January 2024) at 15 Pandora stations located in Northeast Asia (China, South Korea, Japan) from the Pandonia Global Network (PGN). We select data with quality flags of 0, 1, 10, or 11, and with SZA ≤ 70°.

## 3 Reprocessing of GEMS and TROPOMI retrievals to use common vertical profiles

Spectral fitting of satellite solar backscatter observations yields NO$_2$ SCDs, which must be converted to VCDs (the appropriate geophysical quantity) using air mass factors (AMF = SCD/VCD) (Palmer et al., 2001). Standard retrieval algorithms separate the stratospheric and tropospheric portions of the total VCDs to account for the NO$_2$ background in the stratosphere from oxidation of nitrous oxide (N$_2$O) (Bucsela et al., 2013), but this introduces uncertainty related to the stratosphere-troposphere separation (STS) algorithm which may differ between retrievals (Geddes et al., 2018) and leads to ambiguity in the allocation of near-tropopause NO$_2$ such as from lightning and aircraft (Travis et al., 2016; Dang et al., 2023). Since the stratospheric background component is readily predictable from CTMs such as GEOS-Chem (Knowland et al., 2022) and the spatial structure in the NO$_2$ column is mainly from the troposphere, the purpose for separating stratosphere and troposphere is in fact not clear. We focus here on total NO$_2$ VCDs as most useful for scientific applications and to avoid STS errors.

The AMF from the surface to the top of the atmosphere (TOA) can be decomposed into two parts, scattering weights and vertical shape factors, as follows:

$$\mathrm{AMF} = \int_0^{\mathrm{TOA}} w(z)S(z)dz, \tag{1}$$

where $w(z)$ is the scattering weight at altitude $z$ measuring the sensitivity of the instrument to NO$_2$ at altitude $z$ as computed by a radiative transfer model (RTM), and $S(z)$ is the vertical shape factor of normalized number densities obtained from a CTM. Scattering weights for a given wavelength depend on SZA, VZA, relative azimuth angle, surface albedo, and cloud and aerosol optical depths and vertical distributions (Kwon et al., 2019; Hong et al., 2017). The operational GEMS retrieval algorithm uses a look-up-table (LUT) of pre-calculated scattering weights from the VLIDORT RTM at 441 nm (Spurr and Christi, 2014). TROPOMI retrievals use a LUT from the Doubling-Adding KNMI (DAK) RTM at 437.5 nm (Stammes, 2001).

The vertical shape factor $S(z)$ in Eq. (1) describes how NO$_2$ is distributed with altitude as determined by emissions, chemistry, and transport. It must be prescribed in the retrieval as independent information. The L2 GEMS and TROPOMI retrievals use normalized partial column densities simulated for the local scene by GEOS-Chem (Bey et al., 2001; Yang et al., 2023) at 0.25° × 0.3125° resolution and TM5-MP (Williams et al., 2017; Bucsela et al., 2013) at 1° × 1° resolution,



respectively. However, the L2 version 2.0 GEMS product used incorrect GEOS-Chem vertical coordinates. To eliminate differences caused by using different vertical shape factors, we replace the profiles in the GEMS and TROPOMI L2 products with identical GEOS-Chem model profiles extending from the surface to the stratopause. Yang et al. (2023) showed that

GEOS-Chem successfully reproduces the vertical profiles of $NO_2$ and their diurnal variations observed over South Korea in the KORUS-AQ aircraft campaign, supporting the use of GEOS-Chem profiles in a common AMF calculation for GEMS and TROPOMI satellite retrievals in East Asia. We use monthly mean hourly profiles from a 0.25° × 0.3125° resolution simulation with GEOS-Chem version 13.0.0 (DOI: 10.5281/zenodo.4618180). Details on model configuration and emissions can be found in Lee et al. (2024).

Scattering weights are not provided in the GEMS L2 $NO_2$ product, therefore we use scattering weights calculated by VLIDORT using GEMS geometry and atmospheric conditions at 448 nm, which are provided in the GEMS L2 glyoxal product (National Institute of Environmental Research 2020a). The GEMS glyoxal algorithm uses a spectral fitting window of 433.0–461.5 nm (Ha et al., 2024), which sufficiently overlaps with the $NO_2$ fitting window (435–450 nm). The AMF for the GEMS total column can then be calculated using vertical shape factors from GEOS-Chem, following Eq. (1). The

TROPOMI L2 product reports averaging kernels $A(z)$, which normalize the scattering weights to the reported AMF (Eskes and Boersma, 2003).

$$A(z) = \frac{w(z)}{AMF} \qquad (2)$$

The AMF for the TROPOMI total column can then be calculated from the GEOS-Chem vertical shape factors as follows:

$$AMF' = AMF\int_0^{TOA} A(z)S(z)dz, \qquad (3)$$

where AMF is from the L2 product and AMF' is the reprocessed value using GEOS-Chem vertical shape factors. The TROPOMI product reports $A(z)$ for 34 layers, corresponding to the TM5-MP vertical grid, and we interpolate the values to the 47-layer vertical grid of GEOS-Chem. GEMS pixels with AlgorithmQualityFlags > 112, AMFQualityFlags > 64, FinalAlgorithmFlags > 1, and TROPOMI pixels with qa_value < 0.75 are filtered out as per quality control recommendations. We apply area-weighted regridding to the filtered satellite products and use hourly gridded data at 0.25° ×

0.3125° resolution with cloud fraction ≤ 0.3 and SZA ≤ 70° for the remainder of this study.

Figure 1 shows how the reprocessing of AMFs modifies the TROPOMI and GEMS $NO_2$ VCDs compared to the operational L2 products. In what follows, we denote the operational L2 products as "TROPOMI L2" and "GEMS L2", and the products reprocessed with GEOS-Chem vertical profiles as "Reprocessed TROPOMI" and "Reprocessed GEMS". The reprocessing increases TROPOMI in the Northeast Asia source regions including eastern China, South Korea, and Japan. GEMS

decreases over eastern China and increases elsewhere.

Figure 2 compares $NO_2$ VCDs from the L2 and reprocessed products for the ensemble of GEMS and TROPOMI daily data sampled at the overpass time of TROPOMI. For the L2 products we find a negative normalized mean bias (NMB =



$\frac{\sum(GEMS-TROPOMI)}{\sum TROPOMI} \times 100\%$) of −14% in GEMS compared to TROPOMI due to lower background values, but source regions are higher in GEMS. Reprocessing to common prior profiles greatly reduces GEMS-TROPOMI differences except in the background where some differences increase.

Figure 3a compares the mean seasonal variations of Pandora and satellite $NO_2$ VCDs averaged over the 15 Pandora stations in Northeast Asia. The VCDs are maximum in winter and minimum in summer, reflecting the lifetime of $NO_x$ against photochemical oxidation. The operational TROPOMI L2 product has a −16% NMB relative to the Pandora data that is reduced to 7% when reprocessed with GEOS-Chem vertical profiles. The operational GEMS L2 product has a 23% NMB relative to Pandora that is reduced to 7% when reprocessed with GEOS-Chem vertical profiles. The reprocessed TROPOMI and GEMS products are in close agreement, in contrast to the large differences between the TROPOMI L2 and GEMS L2 products, showing that much of the discrepancy in the L2 products stem from different vertical shape factors. In the following section we correct the remaining discrepancy using machine learning.

Figures 3b and 3c show the mean diurnal variations in the warm and cold seasons, comparing GEMS and Pandora. The Pandora data in the cold season increase over the course of the day due to daytime emissions, while the data in the warm season are minimum in early afternoon due to chemical loss (Yang et al., 2024). The operational GEMS L2 data feature a midday maximum in the cold season that is not seen in the Pandora data. Our reprocessed product is more consistent with the diurnal variation observed by Pandora. More detailed comparisons of diurnal variations in Pandora and GEMS are presented by Yang et al. (2024) for Beijing and Seoul.

## 4 Bias correction in GEMS using machine learning

Here we construct a corrected GEMS product by developing an ML model that can predict the differences, Δ(GEMS-TROPOMI), remaining between GEMS and TROPOMI after reprocessing to common vertical profiles. TROPOMI is used as reference because of the greater maturity of its retrieval. The ML model uses as predictors the GEMS $NO_2$ VCD and the GEMS retrieval parameters provided in the L2 product including effective zenith angle (EZA), relative azimuth angle, aerosol optical depth, aerosol layer height, $O_3$ column amount, surface reflectance at 440 nm, single scattering albedo, cloud fraction, and cloud top pressure. EZA combines the geometric effects of SZA and VZA on the AMF, as defined by the geometric AMF = sec(EZA) + 1 in the absence of scattering (Palmer et al., 2001):

$$sec(EZA) = sec(SZA) + sec(VZA) - 1. \qquad (4)$$

TROPOMI observations are for a single time of day but extend off-track to viewing angles ± 50°, so that the collocated dataset covers EZA values ranging up to 75°. This allows us to build an ML model relevant to GEMS observations at different times of day.

We tested five ensemble method algorithms including both bagging (Random Forest, Extra Tree) and boosting (LightGBM, XGBoost, CatBoost) using the Fast and Lightweight AutoML Library (FLAML) (Wang et al., 2021). We separated



collocated Δ(GEMS-TROPOMI) pairs into training (July and October 2022, January and April 2023) and test (rest of the
collocated data for July 2022–June 2023) datasets. The average NO$_2$ VCD from the test dataset is higher than the training
data by 5% but the two datasets display similar spatial distributions. We trained the models to fit 7,489,498 Δ(GEMS-
TROPOMI) pairs (training data) for four months as representative of the four seasons, and found that the LightGBM
algorithm has the best performance. We excluded Δ(GEMS-TROPOMI) data lying outside six times the interquartile range
(6 × IQR; 0.1% of the training data) to avoid contamination by outliers.

We can determine the contribution of each predictor variable to the model prediction using the SHapley Additive
exPlanations (SHAP) analysis with the TreeExplainer method (Lundberg et al., 2020), as shown in Figure 4. The SHAP
value can be interpreted as the relative importance of the predictor variable to the bias correction, where negative SHAP
values indicate low biases in GEMS, and vice versa. Figure 4a shows that the GEMS NO$_2$ VCD contributes to the largest
corrections, followed by the EZA, while variables related to atmospheric scattering and surface reflectance are less
important. Figure 4b shows that the corrections from NO$_2$ VCD and EZA are strongly non-linear. The NO$_2$ VCD drives the
correction of the low bias in the ocean background and the high bias in polluted regions. EZA drives a correction for high
biases at angles exceeding 60°. The dominant corrections from VCD and EZA might be viewed as reflecting a correction
from the SCD, as we would have SCD/VCD = sec(EZA) + 1 in the absence of scattering. However, we found that using
GEMS SCD as a predictor variable was less successful than using VCD.

Figure 5 compares observed and predicted differences for the test data. We conducted a Z-score transform to correct
possible systematic biases associated with the LightGBM algorithm (Belitz and Stackelberg, 2021; Balasus et al., 2023). The
R$^2$ for the ML prediction is 0.51 and the root-mean-square-error (RMSE) is 0.65×10$^{15}$ molecules cm$^{-2}$, which lies within the
estimated single-retrieval errors of GEMS and TROPOMI NO$_2$ of 0.15−2.47×10$^{15}$ molecules cm$^{-2}$ (Kim et al., 2020) and
approximately 0.5×10$^{15}$ molecules cm$^{-2}$ (Van Geffen et al., 2022), respectively.

We produced a corrected GEMS product for the duration of the GEMS L2 version 2.0 record (November 2020 to present)
by subtracting Δ(GEMS-TROPOMI) from the GEMS data previously reprocessed to the GEOS-Chem vertical profiles:

GEMS$_{corrected}$ = GEMS − Δ(GEMS-TROPOMI).                                             (5)

The ML calculation of Δ(GEMS-TROPOMI) requires only GEMS retrieval information and is therefore applied to all
GEMS retrievals, not requiring collocation with TROPOMI.

Figure 6 compares TROPOMI, GEMS, and the corrected GEMS NO$_2$ VCDs for the warm and cold seasons. The corrected
GEMS product increases the ocean background in GEMS by up to 200% and decreases VCDs in Central Asia by up to 40%,
regardless of season. However, corrections to the GEMS product in the polluted regions in Northeast Asia display different
patterns during the warm and cold months. We also see from Figure 2 that the correction successfully reduces remaining
residual differences between GEMS and TROPOMI (NMB = 0%) and increases consistency with the observed variability
from TROPOMI (R$^2$ = 0.72).



Figure 7 shows an enlarged view of the Northeast Asia domain along with observations from 15 Pandora stations. In the warm season, GEMS displays consistent 10% negative biases relative to TROPOMI in eastern parts of China and South Korea (Figure 7d), resulting in an upward correction. The GEMS bias in the cold season is much noisier and tends to be positive, resulting in downward correction.

The effect of the correction on the diurnal profiles observed by GEMS at the Pandora sites is shown in Figure 3b−c. The correction in the cold season decreases GEMS by similar increments for all hours of the day, resulting in no change in the diurnal profile. The corrected GEMS agrees better with Pandora. The correction in the warm season decreases GEMS values only in early morning and late afternoon, modifying the diurnal profile, but the comparison to Pandora is ambiguous. The Pandora data, observing the urban cores, may be less representative of the GEMS observations in summer than in winter

when the $NO_x$ lifetime is longer and winds are stronger (Yang et al., 2024).

## 5 Conclusions

We have presented an improved $NO_2$ vertical column density (VCD) product from the Geostationary Environment Monitoring Spectrometer (GEMS) by calibrating it to TROPOspheric Monitoring Instrument (TROPOMI) with a machine learning (ML) algorithm. A first step was to reprocess both GEMS and TROPOMI datasets to adopt common $NO_2$ vertical

profiles and resulting air mass factors (AMFs) from the GEOS-Chem model. The second step was to correct the residual difference in Δ(GEMS-TROPOMI) with an ML model. The corrected GEMS product preserves the data density of GEMS, providing hourly daytime data over East/South Asia and neighboring oceans, and is consistent with TROPOMI. It is available for the duration of the GEMS record (November 2020 to present).

Reprocessing with a common AMF removed most of the differences between GEMS and TROPOMI operational L2

products. It also resulted in better agreement with the ground-based Pandora observations including for GEMS diurnal profiles. Even after this reprocessing, GEMS displayed low biases compared to TROPOMI in polluted regions of eastern China and South Korea, as well as in the ocean background, and high biases in Central Asia.

We used the LightGBM ML algorithm to correct these remaining biases in GEMS relative to TROPOMI. We trained the ML model to fit collocated Δ(GEMS-TROPOMI) pairs for four months (July and October 2022, January and April 2023) to

GEMS $NO_2$ VCDs and GEMS retrieval parameters. This took advantage of the wide range of TROPOMI viewing angles to train the ML in a manner relevant to GEMS observations at different times of day. The ML was successful in correcting the remaining GEMS differences with TROPOMI. SHAP analysis showed that $NO_2$ VCD and effective zenith angle (EZA) were the predictor variables associated with largest corrections. ML correction increases the ocean background in GEMS by up to 200% and decreases VCDs in Central Asia by up to 40%. GEMS values in $NO_x$ source regions of eastern China and South

Korea increase by ~10% during the warm season but decrease during the cold season, resulting in better agreement with Pandora observations. The GEMS correction in these source regions is similar for all times of day in the cold season, but is largest for early morning and late afternoon in the warm season.



Our corrected GEMS NO$_2$ product is designed to be consistent with the TROPOMI product, supporting the combined use of both datasets for analyses of East Asia air quality including NO$_x$ emissions and chemistry and their diurnal variations. Our

approach of calibrating GEMS NO$_2$ observations to TROPOMI can be extended to other observed species (such as formaldehyde or glyoxal) and to other geostationary satellite instruments including TEMPO over North America and Sentinel-4 over Europe. This would produce consistent datasets across the geostationary air quality constellation with reference to a common TROPOMI calibration for global observing capability.

## Data availability

GEMS L2, TROPOMI L2, and Pandora NO$_2$ products are available online through https://nesc.nier.go.kr/en/html/index.do, https://dataspace.copernicus.eu/explore-data/data-collections/sentinel-data/sentinel-5p, and https://www.pandonia-global-network.org, respectively. The corrected GEMS product from November 2020 to present will be available on Harvard Dataverse upon publication.

## Author contributions

Original draft preparation, data processing, analysis, investigation, and visualization were done by YJO. DJJ and NB contributed to project conceptualization. Review and editing were done by DJJ, NB, HC, RJP, H-AK, and JK. LHY, JP, HL, GTL, and ESH provided additional resources and support in analysis.

## Competing interests

The contact author has declared that none of the authors has any competing interests.

## Financial support


This research was supported by the Samsung Advanced Institute of Technology.

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



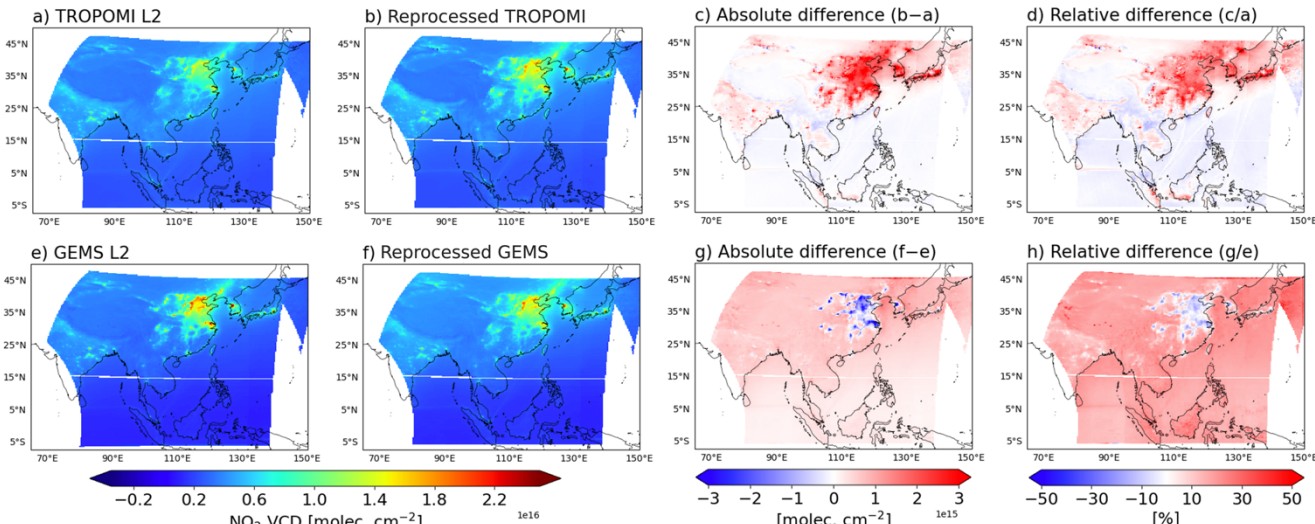

**Figure 1: NO₂ vertical column densities (VCDs) from TROPOMI and GEMS. Values are averages for July 2022−June 2023 sampled at the overpass time of TROPOMI (13:30 local time; LT). The top panels show the TROPOMI operational product (L2), our product reprocessed with GEOS-Chem NO₂ vertical profiles, and the absolute and relative differences between the two. The lower panels show the same for GEMS.**

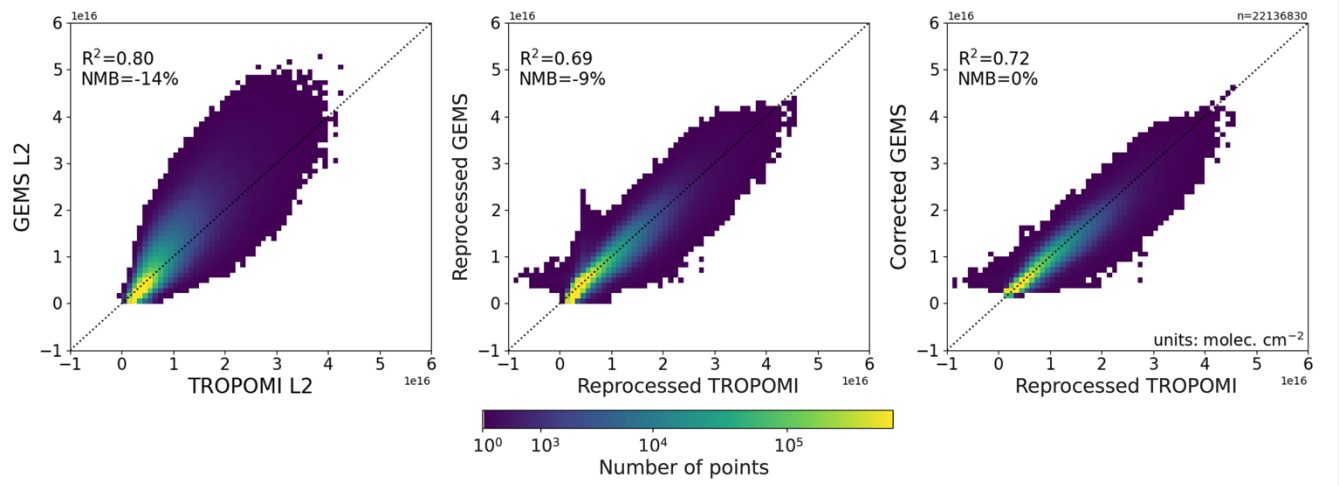

**Figure 2: Scatterplot comparison of NO₂ VCDs from TROPOMI and GEMS. Individual points are daily data for July 2022−June 2023 on the 0.25° × 0.3125° grid sampled at the overpass time of TROPOMI. The left panel compares the operational TROPOMI and GEMS products and the middle panel compares our reprocessed products with GEOS-Chem NO₂ vertical profiles, respectively. The right panel compares the reprocessed TROPOMI product with the GEMS data corrected for residual differences with TROPOMI using machine learning (ML) (Section 4). Colorscale shows density of points. The dashed line indicates the 1:1 line. Coefficient of determination ($R^2$) and normalized mean bias (NMB) are given inset.**



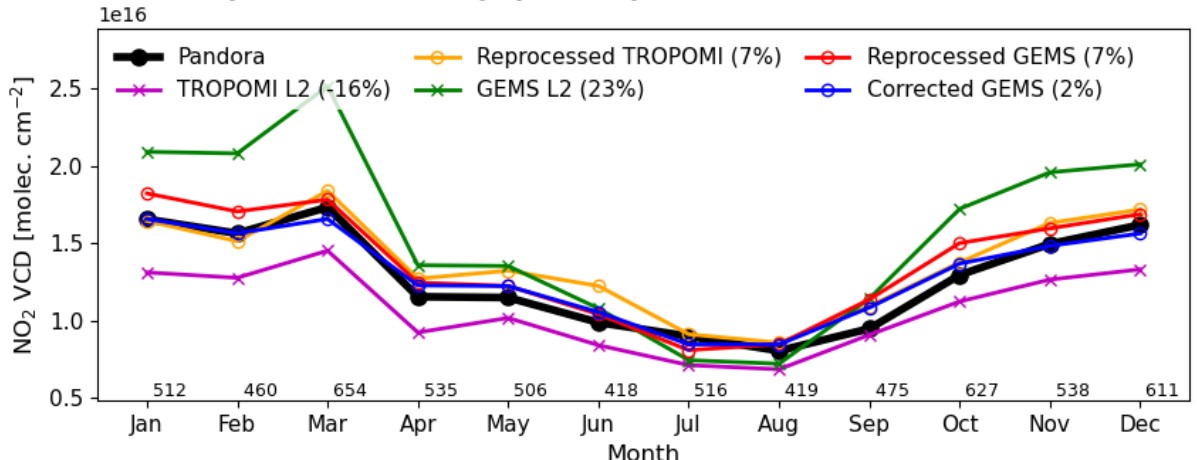

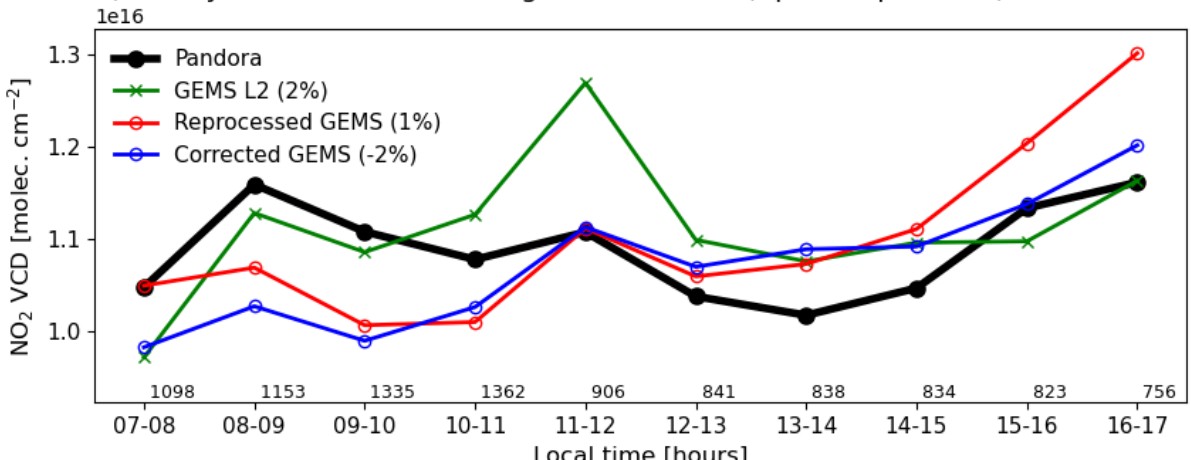

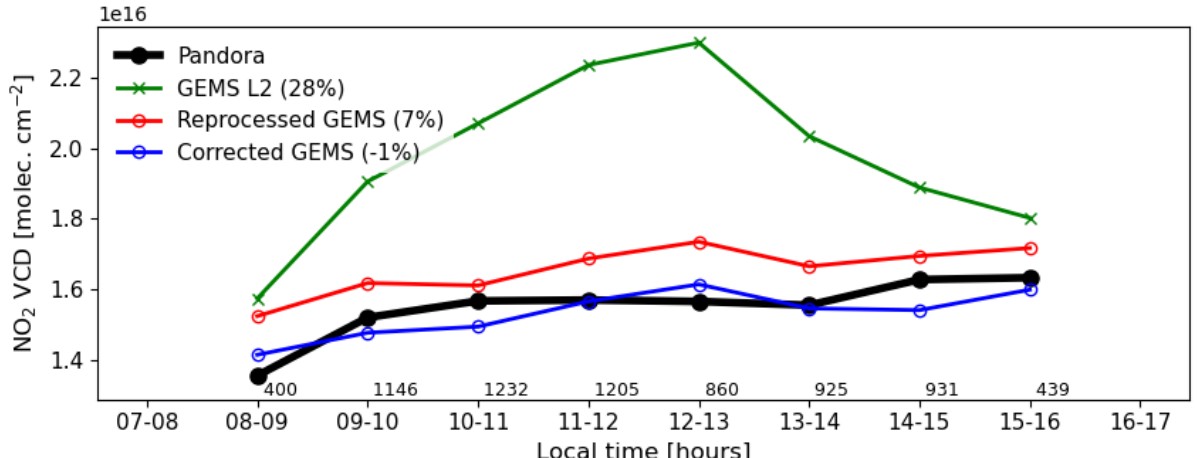





**Figure 3: Seasonal and diurnal variations of NO₂ VCDs from Pandora, TROPOMI, and GEMS averaged at 15 Pandora stations in Northeast Asia over July 2022−June 2023. TROPOMI and GEMS data are shown for the operational L2 products, and the products reprocessed with GEOS-Chem NO₂ vertical profiles. Also shown is the GEMS product corrected for residual differences with TROPOMI using ML (Section 4). Seasonal variations in the top panel are for the TROPOMI overpass time (13:30 LT). Diurnal variations in the middle and bottom panels are only for Pandora and GEMS and are shown for April−September and October−March. NMB relative to Pandora are given inset. The numbers of observations for each month and hour are indicated.**



## a) SHAP analysis of predictor variables

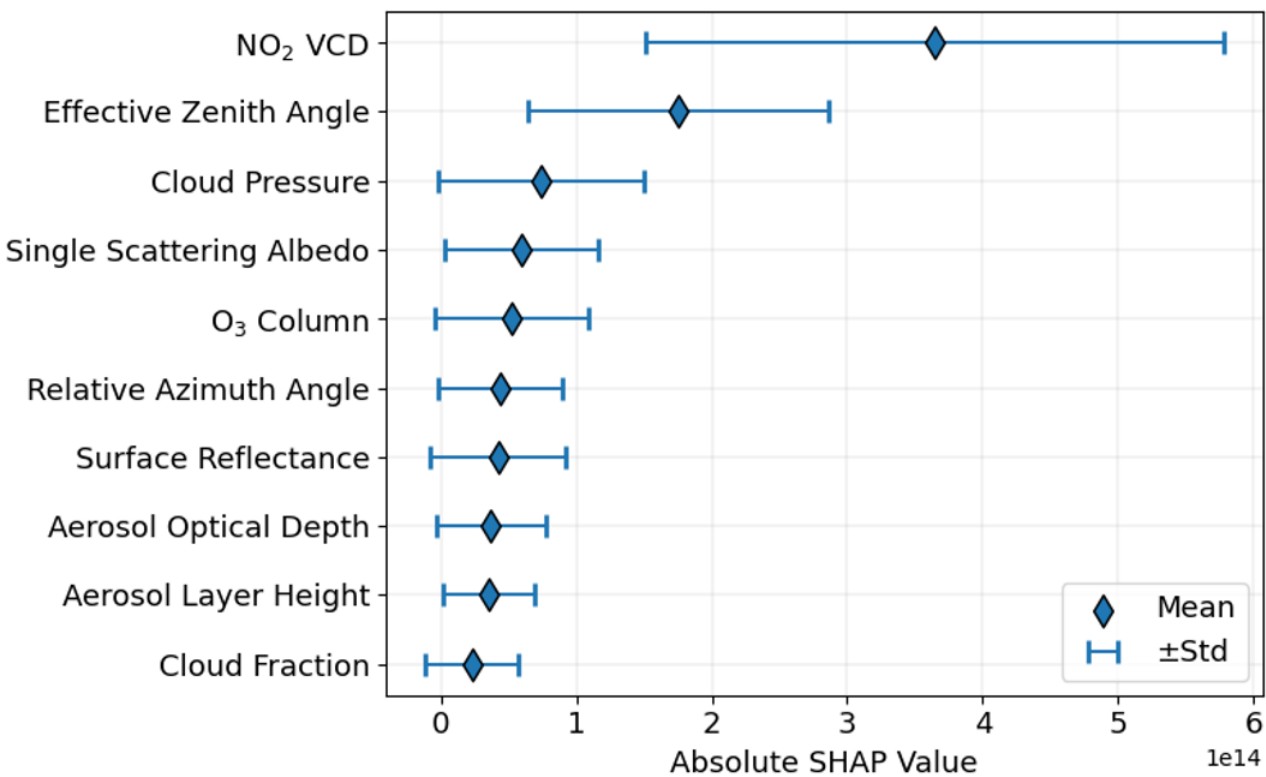

## b) SHAP contribution to Δ(GEMS-TROPOMI)

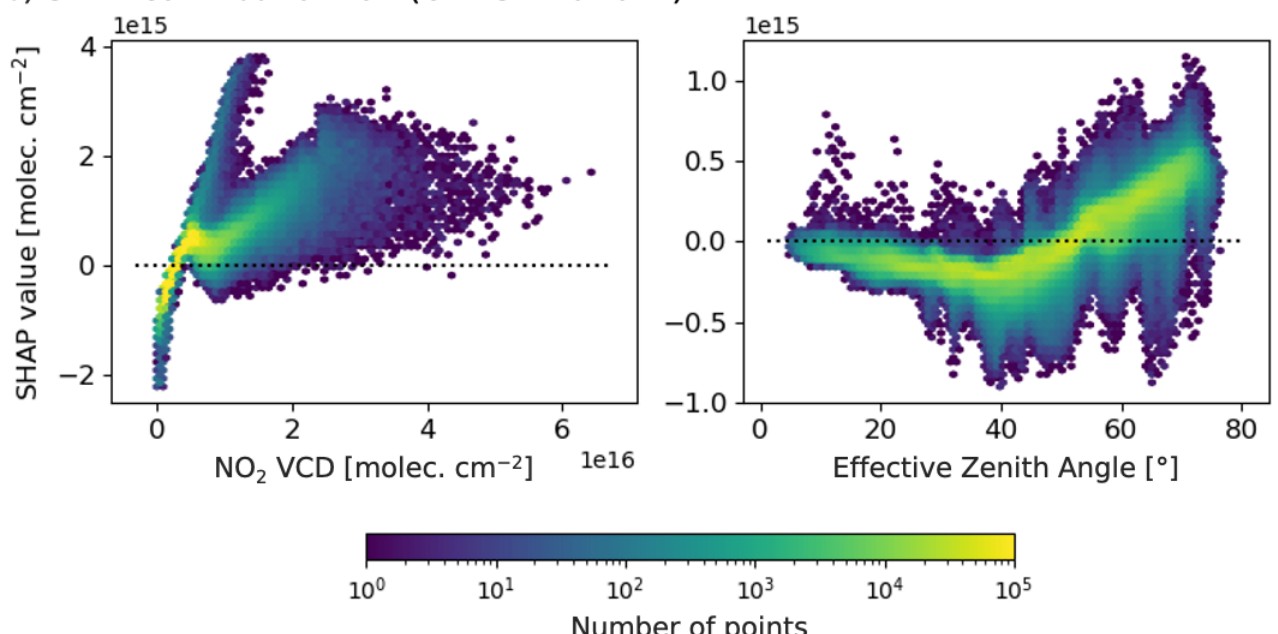



**Figure 4: SHAP analysis for predictors of GEMS-TROPOMI differences, Δ(GEMS-TROPOMI), in the ML training dataset. (a) SHAP analysis results ranking the predictor variables in order of their contributions to the fit. (b) SHAP contribution to Δ(GEMS-TROPOMI) from GEMS NO₂ VCD and effective zenith angle (EZA). Colorscale shows density of points.**

485

**Figure 5: Predicted versus observed Δ(GEMS-TROPOMI) in the test dataset. Colorscale shows density of points. The dashed line indicates the 1:1 line. $R^2$ and root mean square error (RMSE) are given inset.**



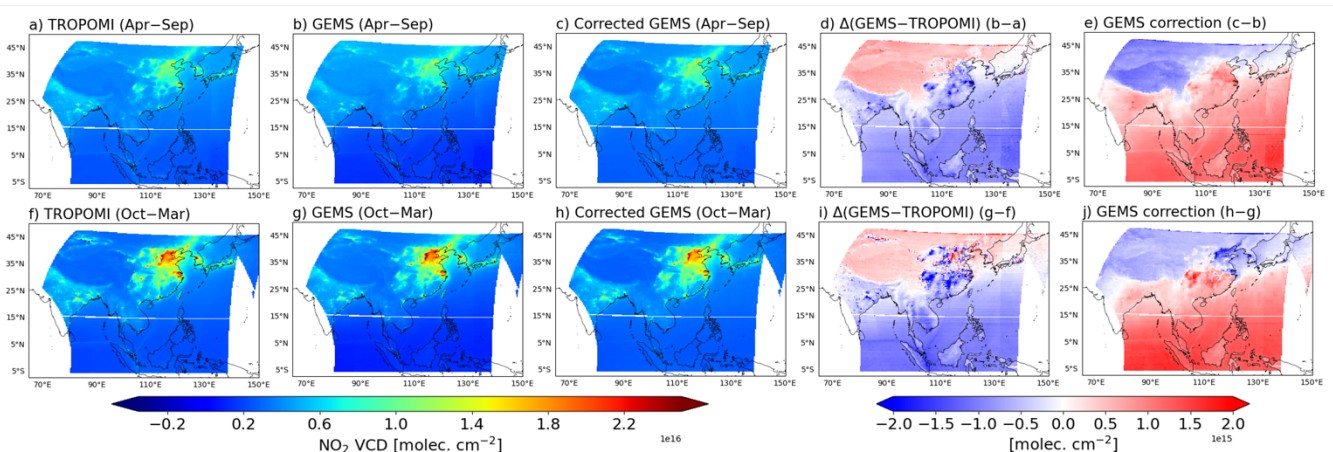

**Figure 6: Comparison of TROPOMI, GEMS, and corrected GEMS NO₂ products in the GEMS scan domain. (a–c) NO₂ VCDs averaged for April–September at the TROPOMI overpass time. The TROPOMI and GEMS data have been reprocessed to common GEOS-Chem vertical profiles for the observation scenes. (d) Δ(GEMS-TROPOMI) for April–September. (e) Correction to the GEMS product for April–September. (f–j) Same as panels (a–e) but for October–March.**

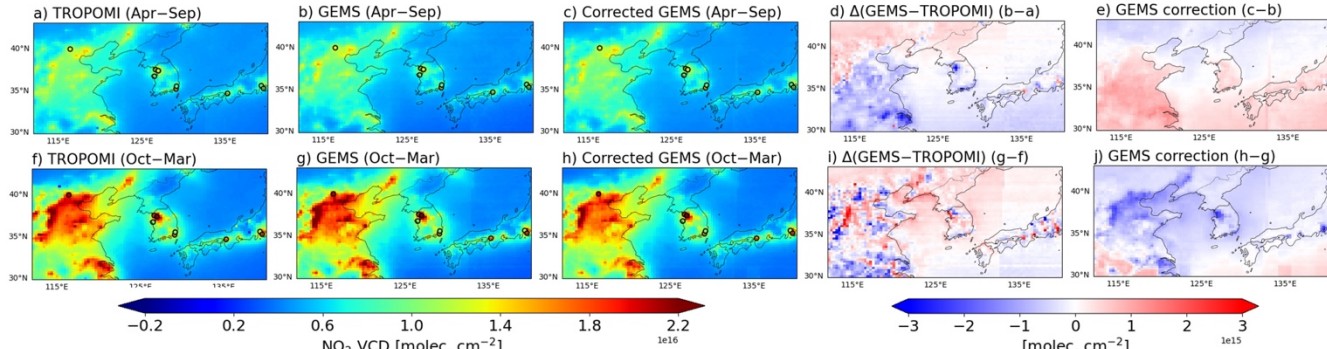

**Figure 7: Same as Figure 6 but in the Northeast Asia domain with Pandora observations shown as circles. Colorscales are different from Figure 6.**