# Peer review of "A bias-corrected GEMS geostationary satellite product for nitrogen dioxide using machine learning to enforce consistency with the TROPOMI satellite instrument"

_EGUsphere, 2024_

## Author Response (AR1)

**Authors' response to comments**

We thank the referees for their thorough review and constructive comments. We have addressed these comments and responses can be found in the colored fonts below.

**Referee #1**

Review of "A bias-corrected GEMS geostationary satellite product for nitrogen dioxide using machine learning to enforce consistency with the TROPOMI satellite instrument" (EGUsphere-2024-393) by Oak et al.

Recommendation: Minor Revision

Summary Statement: This article demonstrates a machine learning model can help to reduce the bias of GEMS geostationary satellite product of nitrogen dioxide compared to the TROPOMI product. This manuscript is well-written and presents a clear and concise approach to obtain bias-corrected GEMS product.

One concern is about the paragraph discussing the SHAP results (lines 195-200). While the contribution of different input variables to the model's performance is an important aspect, I would recommend delaying this discussion until after the performance of the ML model itself has been addressed. This would allow the reader to understand the overall effectiveness of the model before delving into the details of model and specific input variables.

We moved the paragraph to lines 220-229, after evaluating the model's performance, and switched the order of figures accordingly.

Minor comment:

Line 248: It's hard to understand the meaning of "ML correction increases the ocean background" Do you mean that ML correction increases product population over the ocean?

We clarified the sentence to (lines 207-209):

"...GEMS product increases VCDs in the remote ocean background in the southeastern part of the GEMS scan domain by up to 200% and decreases VCDs in Central Asia by up to 40%, regardless of season."

And also in lines 255-256:

"ML correction increases VCDs in the remote ocean regions by up to 200% and decreases VCDs in Central Asia by up to 40%."

**Referee #2**

Oak et al. present an interesting study in which they compare GEMS and TROPOMI NO2 total vertical columns. They find that by recalculating the AMF using consistent GEOS-CHEM profiles for GEMS and TROPOMI, the differences between GEMS and TROPOMI columns are greatly reduced. Furthermore, the comparison with PANDORA data is also improved by this step, both for TROPOMI and GEMS. Finally, they use a ML model to further improve the agreement between GEMS and TROPOMI columns. Their work shows how TROPOMI data can be correctly use as a transfer between the different geostationary instruments.

The results are clearly and honestly presented. It is appreciated that the comments addressed during the quick report have been included. I recommend publication after minor revisions. I would like to read more details/discussion about the points below.

1. It is an interesting result to show that the main reason for the differences between GEMS and TROPOMI NO2 VCD lies in the AMF calculation. The relatively good agreement between the reprocessed columns shows that the NO2 SCD retrieval are consistent. Concerning the GEMS NO2 AMF, since the AK are taken from the GEMS CHOCHO L2 product, we cannot exclude another issue than a wrong use of the GEOS-Chem vertical coordinates.

P6, line 166: "much of the discrepancy in the L2 products stem from different vertical shape factors". Please remind the reader that a large part could also come from an incorrect use of the vertical coordinates in the GEMS NO2 operational product.
We agree on pointing out the issue, which will eventually be corrected in the next operational GEMS product. We added the following sentence (lines 167-169):
*"Incorrect use of vertical coordinates in the L2 version 2.0 GEMS product will be corrected in the next operational product with the GEOS-Chem profiles used here for reprocessing."*

2. It is not shown that the ML model improves the diurnal variation comparison with the PANDORA (mainly from Figure 3). There is no evidence that including the TROPOMI VZA up to 50° actually helps to "build an ML model relevant to GEMS observations at different times of day", as stated p6, line 185, in the abstract and in the conclusions. Please comment on the possibility to further improve the GEMS diurnal variation using ML technique.
Related to this point, it is not clear why the diurnal variation is more affected by the ML model during warm months than during cold months. I expect larger angles during cold months, and therefore a larger correction. Maybe it is because the days are longer during warm days?
We agree that it is unclear that the ML correction improves the diurnal variation comparison to Pandora, however it reduces the biases. We clarified in the conclusion section as follows (line 247):
*"It also resulted in better agreement with the ground-based Pandora observations."*

Despite the fact that we only used collocated data at TROPOMI's overpass time (~13:00) to train the ML model this model produces reasonable data at other times of the day, which was one of our largest concerns on applying it to rest of the GEMS data. This implies that the ML model is valid for EZA values ranging up to 75° so that it can be applied to the full diurnal range of GEMS observations. We clarified this as follows (lines 187-188):
*"This allows us to build an ML model valid for a sufficient range of EZAs, as is necessary for application to GEMS observations over their full diurnal range."*

Pandora or MAX-DOAS observations may serve as reference to further correct biases associated with the diurnal profile retrieved by GEMS. However this may introduce spatial discontinuity to the GEMS data since the ML correction will be limited to only the regions where ground-based observations are available.

The plot below shows that large EZA (> 50°) drives the downward correction during the early morning and late afternoon during the warm season. Since the EZA is always relatively large during the cold season, we see a uniform downward correction for all hours of the day. We added the following to the manuscript (lines 230-237):
*"The correction in the warm season dampens the diurnal variability because EZA varies from 50° at local noon to 65° at 07:00 or 17:00 LT, a range limited by the constant VZA set by latitude. The*

*corrected diurnal variability improves agreement with Pandora in late afternoon while degrading it in early morning. By contrast, the correction in the cold season decreases GEMS by similar increments for all hours of the day, resulting in no change in the diurnal profile but better agreement with Pandora. The lack of diurnal variability in the correction is because EZA varies over only a limited range. The VZA averages 44° for the Northeast Asia Pandora sites, and the SZA ranges from 52° at local noon to 65° at 08:00 or 16:00 LT, so that the EZA varies only from 60° to 68°."*

[Figure]

3. Figure 2: The GEMS NO2 columns seem to be cut for negative values. Is it an effect of the GEMS quality flags? This cutting effect seems to be amplified by the reprocessing and ML correction steps. The correlation is degraded from step 1 to 2. Have you tried to apply an improved quality filtering for GEMS? Or would it make sense to filter negative TROPOMI columns as well?

We did not apply any flags that filter out negative values in both GEMS and TROPOMI, as negative VCD values commonly appear in satellite retrievals and are usually treated as valid data. However we found that there are actually no negative values present in the GEMS total VCDs (although there are negative values in the tropospheric VCDs as a result of subtracting the stratospheric columns), whereas TROPOMI has negative values even in the total VCD product. We revised the manuscript for clarification as follows (line 149):

*"No other filtering is used."*

And in Figure 2 caption (line 481):
*"Negative TROPOMI values reflect noise in the SCD spectral fitting. The GEMS L2 product has no negative values."*

4. Figure 3a: The corrected GEMS columns do agree better with PANDORA than the reprocessed GEMS columns. However, it is not obvious that they agree better with the reprocessed TROPOMI columns (it is the case for Jan/Feb, but not in May or June). It looks like the ML model tend to decrease the GEMS columns but has difficulties to increase them even when there is a negative difference with TROPOMI. Can you comment on this?

Zooming into Figure 3a looking only at TROPOMI and GEMS at the Pandora sites and compare during separate months as shown below, we find that the ML model performs an upwards correction during May to August, and a downwards correction during rest of the months. The downward correction during the colder months deviates the corrected GEMS from the reprocessed TROPOMI (resulting in a low bias of 3.9%), however the reprocessed GEMS product displays a similar bias but in the opposite direction of +3.2% compared to TROPOMI during these months. During the warm months the upward correction (~1%) is indeed smaller than the downward correction (~7%) during September to April. We included additional explanation on Figure 3a as follows (lines 215-217):

*"Comparison of the monthly mean variations of GEMS VCDs at the Pandora sites in Northeast Asia also shows that the ML model performs an upwards correction during May to August, and a larger downwards correction during rest of the months, resulting in an overall low bias of ~5% compared to TROPOMI (Figure 3a)."*

[Figure]

5. Legend of figure 3: It should be mentioned explicitly that all the NO2 columns are total VCD, including the PANDORA columns.
We revised the captions in all figures accordingly.

---

## Author Response (AR2)

**Response to Report #1**

The manuscript by Oak et al. presents a logical and robust approach to bias correcting geostationary satellite NO2 retrievals against TROPOMI and improving performance against PANDORA. The results and method are clearly presented, and the article will be of likely interest to the audience of AMT. The authors' responses to prior reviewer comments are appropriate; I have only a couple very minor suggestions to add.

At L104: "appropriate geophysical quantity" is a bit vague in its meaning. Instead of "appropriate," perhaps "most relevant for comparisons and interpretation" or something else a bit more precise?

Thank you for the suggestion. We revised the phrase to:

*"...converted to VCDs, the geophysical quantity more relevant for column interpretation, using air mass factors…"*

Figure 6: I have a difficult time seeing the Pandora observations overplotted on the map. I did not realize the Pandora obs were there initially; upon reading the text, I referred back to the figure and still have a hard time making the visual comparison to the satellite products at a zoom less than 200%. Perhaps shift panels d, e, i, and j down below the other six, and enlarge the maps that include the Pandora obs? Or otherwise enlarge the overplotted data points?

We enlarged the Pandora data points on each panel so that they're more visible.